# APPLICATION OF BLURRY MODELS FOR SEMANTIC MODELLING OF OBJECT DOMAINS

**Gulnara Yakhyaeva**
Department of Computer Science
Novosibirsk State University
Novosibirsk, Russia
`gul_nara@mail.ru`

## ABSTRACT

Semantic modelling plays an important role in data processing, enabling a deep understanding of information and the development of intelligent systems. One of the methods is a four-level model of knowledge representation including ontological, theoretical, empirical and statistical levels. The problem of incomplete knowledge makes it difficult to describe axioms in an object domain. The paper discusses an approach in which a precedent model (third level) is created based on precedent knowledge and then, through its fuzzification, statistical knowledge (fourth level) is obtained. This probabilistic knowledge is objective. However, in some domains subjective expert estimates may also be used. In such cases, the process starts with the creation of a blurry (fuzzy) model. The paper proposes a mathematical apparatus for reconstructing a set of precedents based on these estimates and describes the properties of blurry models.

## 1 INTRODUCTION

The complexity of AI-based systems has increased significantly in recent times, allowing them to be developed and implemented with minimal human involvement. When decisions made by such systems begin to affect people's lives, there is a need to understand exactly how these decisions are made using AI Vityaev (2023). This emphasises the importance of explainable artificial intelligence against the backdrop of rapid advances in machine learning and AI technologies. Explainability helps developers better understand the behaviour of models, which allows them to identify their flaws and improve their performance. Understanding the principles of algorithms operation and the reasons for their decisions contributes to the formation of users trust in such systems.

One of the ways to develop the trustworthy AI is to use logical-probabilistic methods Vityaev (2020), semantic programming Gumirov (2019); Goncharov (2020); Nechesov (2023) and construction of semantic models of object domains Palchunov (2022a;b).

A four-level knowledge representation model was proposed in order to formalise the object domain in a set-theoretic way Naydanov C. (2015); Palchunov D. (2016). This model consists of the following levels: ontological knowledge, general theoretical knowledge, individual empirical knowledge and statistical knowledge. Ontological knowledge is defined as reusable knowledge that remains constant when the knowledge model is altered within the same domain. The second level is represented by general theoretical knowledge, which contains expert knowledge about the object domain. The third level consists of individual object domain precedents, and finally, the fourth level includes statistics and probabilistic knowledge, which is generated as a result of statistical analyses of information extracted from individual precedents.

In model-theoretic formalisation, ontological knowledge, in particular, specifies the signature and the basic set of the model describing the object domain Yakhyaeva (2014). A complete description of the first and second levels of the knowledge representation model implies the formation of a set of axioms of the object domain, i.e., statements that will be true for all precedents of the object domain described at the third level.

Often, in practice, due to incomplete knowledge, it is difficult to describe a set of all axioms of an object domain at once. This paper proposes an approach that involves the construction of a precedent model of the object domain, informed by accumulated knowledge about precedents. The subsequent fuzzification of the precedent model results in the acquisition of statistical or probabilistic knowledge about the object domain, corresponding to the fourth level of the knowledge representation model. The subsequent consideration of the theory of the obtained fuzzification will result in the description of the class of object domain axioms, thereby completing the formation of the first and second levels of the knowledge representation model.

The probabilistic knowledge obtained from the fuzzification of the precedent model is objective. However, in some domains subjective assessments of experts can be used as well. In such cases modelling starts from the fourth level - creation of a blurry (fuzzy) model of object domain. The paper presents a mathematical apparatus that allows to reconstruct a set of precedents on the basis of this blurry model.

## 2 Sigma-algebra defined on a class of algebraic systems

In this paper, the subject of predicate logic without equality is considered. The set of all sentences of signature $\sigma$ is denoted by $S(\sigma)$, and the set of all atomic sentences of signature $\sigma$ is denoted by $S_a(\sigma)$. The semantic equivalence relation $\sim$ is introduced on the set of sentences $S(\sigma)$ in the standard way. The Lindenbaum-Tarski algebra $\langle S(\sigma)/_\sim; \vee, \&, \neg, \top, \bot \rangle$ denoted by $\mathbb{S}(\sigma)$.

Let $\mathfrak{A}$ be an object domain described by an algebraic system, denoted by $\mathfrak{A} = \langle A, \sigma \rangle$. More precisely, each concrete precedent of this object domain defines its own algebraic system, i.e. its own evaluation of the signature symbols of the given object domain on the basic set. For simplicity, it will be assumed that all algebraic systems formally describing cases of a given object domain have one basic set, i.e. with precision up to reification, the same set $\{c_a^{\mathfrak{A}} \mid a \in A\}$. A signature $\sigma$ is defined as a set of concepts whose language describes a given object domain. It is assumed that all cases of the object domain have the same signature. It is further assumed that $\sigma \cap \{c_a \mid a \in A\} = \emptyset$ and denoted by $\sigma_A = \sigma \cup \{c_a \mid a \in A\}$.

Next, we denote by $\Omega(\sigma_A)$ the sigma-algebra generated by the set $\{K_\varphi \mid \varphi \in S(\sigma_A)\}$, that is, the minimal algebra containing all elements of the set $\{K_\varphi \mid \varphi \in S(\sigma_A)\}$ and closed under complementation, countable unions, and countable intersections of elements from $\{K_\varphi \mid \varphi \in S(\sigma_A)\}$.

It is noteworthy that if the considered domain is finite, i.e., the set of all atomic sentences $S_a(\sigma_A)$ is finite, then the sigma-algebra $\Omega(\sigma_A)$ is also finite and isomorphic to the class $2^{K(\sigma,A)}$ of all subclasses of the class $K(\sigma, A)$.

On the other hand, if the set $S_a(\sigma_A)$ is infinite, then we have $\|\Omega(\sigma_A)\| = \|S_a(\sigma_A)\|$, while at the same time $\|K(\sigma, A)\| > \|S_a(\sigma_A)\|$. Furthermore, in this case, for each quantifier-free sentences $\varphi \in S(\sigma_A)$, the class $K_\varphi$ is equivalent with the class $K(\sigma, A)$.

Thus, for example, if the set of atomic sentences $S_a(\sigma_A)$ is countable, then the class $K(\sigma, A)$ is continuous and the sigma-algebra $\Omega(\sigma_A)$ contains a countable number of classes of algebraic systems of continuous power.

**Proposition 1** *Let the set of all atomic sentences $S_a(\sigma_A)$ be countable. Then the sigma-algebra $\Omega(\sigma_A)$ contains all finite, co-finite, countable and co-countable subclasses of the class of all algebraic systems $K(\sigma, A)$.*

Proof. Let $S_a(\sigma_A) = \{\varphi_1, \varphi_2, ...\}$. For any atomic sentence $\varphi_i \in S_a(\sigma_A)$ and for any algebraic system $\mathfrak{A} \in K(\sigma, A)$ we introduce the notation

$$\varphi_i^{\mathfrak{A}} = \begin{cases} \varphi_i & \mathfrak{A} \models \varphi_i; \\ \neg\varphi_i & \mathfrak{A} \not\models \varphi_i. \end{cases}$$

Then we get

$$\{\mathfrak{A}\} = \bigcap_{i \in N} K_{\varphi_i^{\mathfrak{A}}}.$$

And since the sigma algebra $\Omega(\sigma_A)$ is closed with respect to countable intersections, then $\{\mathfrak{A}\} \in \Omega(\sigma_A)$, i.e., it contains all one-element classes.

Furthermore, due to the closure of the sigma-algebra under complementation and countable unions, we will obtain the statement of the Proposition.

$\square$

## 3 THE PRECEDENT MODEL AND ITS FUZZIFICATION

In general, not every algebraic system $\mathfrak{A} \in K(\sigma, A)$ can be considered a precedent for a given object domain. This is because, as previously mentioned, all axioms formalized at the first and second levels of knowledge representation must be satisfied for precedents in the object domain. Thus, when modeling the object domain, a subclass $E \subseteq K(\sigma, A)$ of precedents is distinguished, for which a Precedent Model of the given object domain is constructed Yakhyaeva (2021).

**Definition 1** *Let $E \subseteq K(\sigma, A)$. Let us call the ordered triple (generated by set $E$) the precedent model $\mathfrak{A}_E \rightleftharpoons \langle A, \sigma, \tau_E \rangle$, where $\tau_E : S(\sigma_A) \to \wp(E)$, if for any $\varphi \in S(\sigma_A)$ we have*

$$\tau_E(\varphi) = \{\mathfrak{A} \in E \mid \mathfrak{A} \models \varphi\}.$$

Thus, in the precedent model, each sentence of signature $\sigma_A$ corresponds to a set of precedents on which this sentence is satisfied.

The precedent model establishes the semantic description of the object domain and constitutes the third level of the knowledge representation model. In order to progress to the fourth level of the knowledge representation model, i.e. to obtain the quantitative characteristics of the object domain events, it is necessary to fuzzify this precedent model Yakhyaeva (2009). To this end, it is required to introduce a counting-additive measure defined on the class of algebraic systems. $K(\sigma, A)$.

**Definition 2** *The mapping $\nu : \rho(K(\sigma, A)) \to [0, \infty]$ is defined as a countably additive measure, i.e., it satisfies the following properties:*

*(M1)* $\nu(\emptyset) = 0$;

*(M2) (monotonicity property) For any two classes of models $K_1, K_2 \in K(\sigma, A)$ it follows that*

$$K_1 \subseteq K_2 \Rightarrow \nu(K_1) \le \nu(K_2);$$

*(M3) (countable additivity property) For any countable sequence of pairwise non-intersecting classes of models $K_1, K_2, ...$ it follows that*

$$\nu(\bigcup_{i \in N} K_i) = \sum_{i \in N} \nu(K_i).$$

We will say that the measure $\nu : \rho(K(\sigma, A)) \to [0, \infty]$ is defined on the set of precedents $E \subseteq K(\sigma, A)$ if $0 < \nu(E) < \infty$.

**Definition 3** *An ordered triple $Fuz(\mathfrak{A}_E, \nu) = \langle A, \sigma, \mu \rangle$ is called a fuzzification of the precedent model $\mathfrak{A}_E$ by a measure $\nu$ defined on the precedent class $E$ if the mapping $\mu : S(\sigma_A) \to [0, 1]$ satisfies the condition*

$$\mu(\varphi) = \frac{\nu(\tau_E(\varphi))}{\nu(E)}, \text{ for any } \varphi \in S(\sigma_A).$$

For any precedent $\mathfrak{A} \in E$, the measure $\nu(\{\mathfrak{A}\})$ can be interpreted as the 'significance' of the precedent $\mathfrak{A}$ for a given object domain. Thus, if $\nu(\{\mathfrak{A}\}) = 0$, then the precedent $\mathfrak{A}$ is 'irrelevant' to the object domain and can be removed from consideration.

**Proposition 2** *Consider a measure $\nu$ defined on the precedence class $E$. Let $K_0 \subset K(A, \sigma)$ be the model class of measure zero. Then*

$$Fuz(\mathfrak{A}_E, \nu) = Fuz(\mathfrak{A}_{E \setminus K_0}, \nu).$$

Proof. Let $E_1 = E \setminus K_0$ and $E_2 = E \cap K_0$. It is obvious that $E_1 \cap E_2 = \emptyset$ and $E_1 \cup E_2 = E$.

Then, on the one hand, for any sentence $\varphi \in S(\sigma_A)$ we have $\tau_E(\varphi) = \tau_{E_1}(\varphi) \cup \tau_{E_2}(\varphi)$. And on the other hand, $\tau_{E_1}(\varphi) \cap \tau_{E_2}(\varphi) = \emptyset$. Hence, by the properties of the measure $\nu$ we obtain

$$\nu\big(\tau_E(\varphi)\big) = \nu\big(\tau_{E_1}(\varphi)\big) + \nu\big(\tau_{E_2}(\varphi)\big)$$

Since $\tau_{E_2}(\varphi) \subseteq K_0$, by the properties of the measure $\nu$ we have $\nu\big(\tau_{E_2}(\varphi)\big) = 0$. Hence, $\nu\big(\tau_E(\varphi)\big) = \nu\big(\tau_{E_1}(\varphi)\big)$.

By anological reasoning we obtain that $\nu(E) = \nu(E_1)$. Hence, for any sentence $\varphi \in S(\sigma_A)$ we have

$$\frac{\nu(\tau_E(\varphi))}{\nu(E)} = \frac{\nu(\tau_{E_1}(\varphi))}{\nu(E_1)}.$$

$\square$

We will say that the measure $\nu : \rho(K(\sigma, A)) \to [0, \infty]$ is everywhere defined on the set of precedents $E \subseteq K(\sigma, A)$ if $\nu(E') \neq 0$ holds for any proper subclass of precedents $E' \subseteq E$.

**Theorem 1** *Consider the fuzzification $Fuz(\mathfrak{A}_E, \nu)$ of the precedent model $\mathfrak{A}_E$ using a measure $\nu$ defined on the precedent class $E$. Then the evaluation $\mu : S(\sigma_A) \to [0, 1]$ is a probabilistic measure defined on the Lidenbaum-Tarski algebra $\mathbb{S}(\sigma_A)$, i.e., the following conditions are satisfied:*

*(A1) $\mu(\top) = 1$ and $\mu(\bot) = 0$.*

*(A2) (monotonicity property) For any sentence $\varphi, \psi \in S(\sigma_A)$ the following condition holds*

$$\varphi \sim \varphi \& \psi \Rightarrow \mu(\varphi) \leq \mu(\psi).$$

*(A3) (countable additivity property) For any countable sequence of sentences $\varphi_1, \varphi_2, \ldots \in S(\sigma_A)$ if $\mu(\varphi_i \& \varphi_j) = 0$ for any $i, j \in N (i \neq j)$, then it follows*

$$\mu(\bigvee_{i \in N} \varphi_i) = \sum_{i \in N} \mu(\varphi_i).$$

*(A4) (equivalence property) For any sentence $\varphi, \psi \in S(\sigma_A)$ the following condition holds*

$$\varphi \sim \psi \Rightarrow \mu(\varphi) = \mu(\psi).$$

The proof of the Theorem follows directly from the properties of the measure $\nu$.

## 4 BLURRY MODEL OF THE OBJECT DOMAIN

Often, when modeling a object domain, we may initially lack knowledge about the class of precedents for that domain. However, when describing certain object domains, subjective assessments from experts can be used Yakhyaeva G.E. (2023). In this case, the construction of a knowledge representation model begins at the fourth level, namely with the development of a Blurry (fuzzy) Model of the object domain.

**Definition 4** *The triple $\mathfrak{A}_\mu = \langle A, \sigma, \mu \rangle$ will be called a blurry model if the evaluation $\mu : S(\sigma_A) \to [0, 1]$ is a probability measure defined on the Lidenbaum-Tarski algebra $\mathbb{S}(\sigma_A)$ (i.e. the properties (A1)-(A4) are satisfied).*

Note that if the measure $\mu$ describing the blurry model $\mathfrak{A}_\mu$ is trivial (i.e., a mapping to the two-element set $\{0, 1\}$), then the model $\mathfrak{A}_\mu$ is a (classical) model of predicate logic. Such models will be called crisp models of the $\sigma$ signature later in this paper.

Also note that the equivalence property (A4) guarantees the satisfaction of all laws of classical logic on the blurry model, i.e., the notion of a blurry model is a conservative extension of the notion of an crisp model Yakhyaeva (2023).

**Proposition 3** *Let $\mathfrak{A}_\mu = \langle A, \sigma, \mu \rangle$ be a blurry model. Then for any sentences $\varphi, \psi \in S(\sigma_A)$ we have*

1. $\mu(\neg\varphi) = 1 - \mu(\varphi)$;

2. $\mu(\varphi \& \psi) \in \left[ \max\left\{0; \mu(\varphi) + \mu(\psi) - 1\right\}; \min\left\{\mu(\varphi); \mu(\psi)\right\} \right]$;

3. $\mu(\varphi \vee \psi) \in \left[ \max\left\{\mu(\varphi); \mu(\psi)\right\}; \min\left\{1; \mu(\varphi) + \mu(\psi)\right\} \right]$.

Proof. From property (A3) of the measure $\mu$ we can directly obtain that for any prepositions $\varphi, \psi \in S(\sigma_A)$ the following property is satisfied

(A3′) For any sentences $\varphi, \psi \in S(\sigma_A)$ the following condition holds

$$\mu(\varphi \vee \psi) = \mu(\varphi) + \mu(\psi) - \mu(\varphi \& \psi).$$

We'll use it to prove the Proposition.

(1) For any sentence $\varphi \in S(\sigma_A)$ by properties (A4) and (A1) we have

$$\mu(\varphi \vee \neg\varphi) = \mu(\top) = 1 \text{ and } \mu(\varphi \& \neg\varphi) = \mu(\bot) = 0.$$

Hence, by the property (A3′) we get $\mu(\neg\varphi) = 1 - \mu(\varphi)$.

(2) On the one hand, by property (A2), for any sentences $\varphi, \psi \in S(\sigma_A)$ we have $\mu(\varphi \& \psi) \leq \mu(\varphi)$ and $\mu(\varphi \& \psi) \leq \mu(\psi)$. Hence, $\mu(\varphi \& \psi) \leq \min\left\{\mu(\varphi); \mu(\psi)\right\}$.

On the other hand, for any sentences $\varphi, \psi \in S(\sigma_A)$, by properties (A1) and (A2) we have $\mu(\varphi \vee \psi) \leq 1$. Hence, by property (A3′) we get $\mu(\varphi) + \mu(\psi) - \mu(\varphi \& \psi) \leq 1$. Hence, $\mu(\varphi \& \psi) \geq \mu(\varphi) + \mu(\psi) - 1$. And, by property (A1), we get $\mu(\varphi \& \psi) \geq \max\left\{0; \mu(\varphi) + \mu(\psi) - 1\right\}$.

(3) Proved in the same way.

$\square$

**Corollary 1** *Let $\mathfrak{A}_\mu = \langle A, \sigma, \mu \rangle$ be a blurry model. Then for any sentences $\varphi_1, ..., \varphi_n \in S(\sigma_A)$ we have*

1. $\mu(\varphi \& ... \& \varphi_n) \in \left[ \max\left\{0; \sum_{i=1}^{n} \mu(\varphi_1) + n - 1\right\}; \min\left\{\mu(\varphi_i) \mid i = \overline{1, n}\right\} \right]$;

2. $\mu(\varphi_n \vee ... \vee \varphi_n) \in \left[ \max\left\{\mu(\varphi_i) \mid i = \overline{1, n}\right\}; \min\left\{1; \sum_{i=1}^{n} \mu(\varphi_1)\right\} \right]$.

The proof follows directly from the Proposition 3.

**Corollary 2** *Let $\mathfrak{A}_\mu = \langle A, \sigma, \mu \rangle$ be an infinite blurry model. Then for any formula $\varphi(x) \in F(\sigma_A)$ we have*

1. $\mu(\forall x \varphi(x)) \in \left[ 0; \ \min\left\{\mu\left(\varphi(a)\right) \mid a \in A\right\} \right]$;

2. $\mu(\exists x \varphi(x)) \in \left[ \max\left\{\mu\left(\varphi(a)\right) \mid a \in A\right\}; \ 1 \right]$.

The proof follows directly from the Corollary 1.

In classical model theory, the atomic diagram of a model is a subset of the set $S_a(\sigma)$ of all atomic sentences of a given signature. Since we are dealing with models in which all sentences have an evaluative characteristic (fuzzy estimation), the atomic diagram of a blurry model will be understood as a fuzzy subset of the set $S_a(\sigma)$ whose membership function coincides with the measure $\mu$ defining this model.

Let us give a formal definition of this notion.

**Definition 5** *Let $\mathfrak{A}_\mu = \langle A, \sigma, \mu \rangle$ be a blurry model. Then the sets of ordered pairs*

$$AD(\mathfrak{A}_\mu) = \big\{ (\varphi, \mu(\varphi)) \ \ \varphi \in S_a(\sigma_A) \big\},$$

$$PD(\mathfrak{A}_\mu) = \big\{ (\varphi, \mu(\varphi)) \mid \exists n \in N : \varphi = \varphi_1 \& ... \& \varphi_n, \varphi_i \in S_a(\sigma_A) \big\}$$

*let us call respectively the atomic diagram and the positive diagram of the model $\mathfrak{A}_\mu$ .*

As a consequence of Proposition 3, it can be deduced that an atomic diagram does not necessarily imply a blurry model. This means that it is possible to construct non-equivalent blurry models that possess equivalent atomic diagrams.

**Theorem 2** *If the blurry model $\mathfrak{A}_\mu = \langle A, \sigma, \mu \rangle$ is finite or countable, then it is uniquely given by its positive diagram.*

Proof. We need to show that for any sentence $\psi \in S(\sigma_A)$ its truth value $\mu(\psi)$ is uniquely determined by the truth values of positive conjuncts from the positive diagram $PD(\mathfrak{A}_\mu)$.

Case 1. Let the sentence $\psi \in S(\sigma_A)$ be a conjunct, i.e., it has the following form

$$\psi = \varphi_1 \ \& \ ... \ \& \ \varphi_k \ \& \ \neg\varphi_{k+1} \ \& \ ... \ \& \ \neg\varphi_{k+l},$$

where $\varphi_i \in S_a(\sigma_A)$ $(i = \overline{1, k+l})$.

We will prove by induction on the number $l$ of atomic sentences entering with negation into the conjunct $\psi$ .

Let $l = 0$, then the conjunct $\psi$ is positive and its truth value on the model $\mathfrak{A}_\mu$ is given by the positive diagram $PD(\mathfrak{A}_\mu)$.

Suppose now that for any conjunct containing $l - 1$ atomic sentences with negation the statement of the Theorem is true. Let us prove that the statement of the Theorem is also true for the conjunct $\psi$.

From the property (A3$'$) it follows that

$$\mu(\psi) = \mu(\varphi_1 \& ... \& \varphi_k \& \neg\varphi_{k+1} \& ... \& \neg\varphi_{k+l-1}) - \mu(\varphi_1 \& ... \& \varphi_k \& \neg\varphi_{k+1} \& ... \& \neg\varphi_{k+l_1} \& \varphi_{k+l}).$$

Since both conjuncts on the right-hand side of the equality contain exactly $l - 1$ atomic sentences with negation, by induction, their truth values on the model $\mathfrak{A}_\mu$ are uniquely determined by the truth values of positive conjuncts from the positive diagram $PD(\mathfrak{A}_\mu)$. It can thus be concluded that the truth value of the conjunct $\psi$ is also uniquely determined.

Case 2. Let the sentence $\psi \in S(\sigma_A)$ be quantifier-free. Then it is equivalent to some sentence $\psi'$ in SDNF form. Moreover, it follows from property (A4) that $\mu(\psi) = \mu(\psi')$.

Let $\psi' = \omega_1 \vee ... \vee \omega_s$, where $\omega_i$ are some conjuncts. Then, by property (A3$'$), it follows that

$$\mu(\psi') = \mu(\omega_1) + ... + \mu(\omega_s).$$

Since it has already been proved that the truth values of all conjuncts $\omega_i$ on the model $\mathfrak{A}_\mu$ are uniquely determined by the truth values of the positive conjuncts from the positive diagram $PD(\mathfrak{A}_\mu)$, the truth value of the sentence $\psi$ is also uniquely determined.

Case 3. Let the sentence $\psi \in S(\sigma_A)$ be a quantifier. By the condition of the Theorem, the basic set of the blurry model $\mathfrak{A}_\mu$ is either finite or countable. If it is finite, then the proposition $\psi$ is equivalent to some quantifier-free proposition, and hence its truth value on the model $\mathfrak{A}_\mu$ is uniquely determined by the truth values of positive conjuncts from the positive diagram $PD(\mathfrak{A}_\mu)$.

We will consider the case when the basis set of the blurry model $\mathfrak{A}_\mu$ is countable, i.e., $A = \{a_1, a_2, ...\}$.

The proposition $\psi$ is equivalent to some proposition $\psi'$ in the prevalued normal form. Furthermore, it follows from property (A4) that $\mu(\psi) = \mu(\psi')$.

Let $\psi' = Q_1 x_1 ... Q_n x_n \xi(x_1, ..., x_n)$, where $Q_i \in \{\forall, \exists\}$ and $\xi(x_1, ..., x_n)$ is a quantum-free formula. We will prove by induction on the number $n$ of quantifier changes included in the sentence $\psi'$.

If $n = 0$, then the sentence $\psi'$ is either $\forall$-formula or $\exists$-formula.

Consider the case when the sentence $\psi'$ is a $\forall$-formula, i.e. $\psi' = \forall x_1...\forall x_n \xi(x_1, ..., x_n)$, where $\xi(x_1, ..., x_n)$ is a quantifier-free formula. Let us construct a countable number of sentences as follows:

$$\begin{array}{rcl}
\omega_0 & = & \xi(a_{c_n^1(0)}, ..., a_{c_n^n(0)}); \\
\omega_1 & = & \xi(a_{c_n^1(0)}, ..., a_{c_n^n(0)}) \mathrel{\&} \psi''(a_{c_n^1(1)}, ..., a_{c_n^n(1)}); \\
\omega_2 & = & \xi(a_{c_n^1(0)}, ..., a_{c_n^n(0)}) \mathrel{\&} \psi''(a_{c_n^1(1)}, ..., a_{c_n^n(1)}) \mathrel{\&} \psi''(a_{c_n^1(2)}, ..., a_{c_n^n(2)}); \\
& ... &
\end{array}$$

where $c_n^k$ is a Cantor functions giving the value of the $k$-th coordinate of a tuple of natural numbers of length $n$.

Obviously

$$\psi' = \forall x_1...\forall x_n \xi(x_1, ..., x_n) = \lim_{n \to \infty} \omega_n.$$

By property (A2) of the probability measure $\mu$, we obtain

$$\mu(\omega_1) \geq \mu(\omega_2) \geq \mu(\omega_3) \geq ....$$

Due to the boundedness of this sequence of numbers, it has a limit. Hence,

$$\mu(\psi') = \lim_{n \to \infty} \mu(\omega_n).$$

Thus, the truth value of sentence $\psi$ is uniquely determined by the truth values of sentences $\omega_i$. And by virtue of the fact that all sentences $\omega_i$ are quantifier-free, we obtain the statement of the Theorem.

The case when the proposition $\psi'$ is a $\exists$-formula is proved similarly.

Let us now turn to the induction step. Suppose that for any sentence containing $n-1$ permutations of quantifiers the statement is true. Let us prove that the Theorem is also true for a sentence containing $n$ permutations of quantifiers.

Let us consider the case when the sentence $\psi'$ has the form

$$\psi' = \forall x_1...\forall x_k \exists x_{k+1} Q_{k+2} x_{k+2}...Q_n x_n \xi(x_1, ..., x_n),$$

where $\xi(x_1, ..., x_n)$ is a quantum-free formula.

By analogy with the previous case, construct a countable number of sentences as follows:

$$\phi_s = \bigwedge_{i=0}^{s} \exists x_{k+1} Q_{k+2} x_{k+2}...Q_n x_n \xi(a_{c_k^1(i)}, ..., a_{c_k^k(i)}, x_{k+1}, ..., x_n), \; where \; s \in N.$$

Then, by similar reasoning, we obtain that

$$\mu(\psi') = \lim_{n \to \infty} \mu(\phi_n).$$

Obviously, the sentence $\phi_0$ is in a prenex normal form and has $n-1$ quantifier changes. It is also not difficult to show that all other $\phi_i$ can be reduced to a prenex normal form containing exactly $n-1$ quantifier changes. Thus, by induction, we obtain that every sentence $\phi_i$ is uniquely defined by the truth values of the positive conjuncts from the positive diagram $PD(\mathfrak{A}_\mu)$. Hence, the statement of the Theorem is true for the sentence $\psi$ as well.

$\square$

## 5 Defuzzification Theorem for a Blurry Model

As previously mentioned, the construction of a blurry model of the object domain constitutes the fourth level of knowledge representation, with the third level remaining unformalised. The question of whether it is possible to reconstruct emperechi knowledge (i.e. to reconstruct the set of precedents of the object domain) from the available evaluative knowledge (i.e. the blurry model) arises. The answer to this question is provided by the Defuzzification Theorem, which is proven below.

In order to prove the Defuzzification Theorem, it is necessary to introduce the notion of the elementary theory of the blurry model.

**Definition 6** *Let* $\mathfrak{A}_\mu = \langle A, \sigma, \mu \rangle$ *be a blurry model. Then the set of sentences*
$$Th(\mathfrak{A}_\mu) = \{\varphi \in S(\sigma_A) \mid \mu(\varphi) = 1\},$$
*we will call the theory of the model* $\mathfrak{A}_\mu$.

Using properties (A1)-(A4) it is easy to show that the theory $Th(\mathfrak{A}_\mu)$ of any blurry model $\mathfrak{A}_\mu$ is a filter of the Lindenbaum-Tarski algebra $\mathbb{S}(\sigma)$.

**Theorem 3 (about defuzzification)** *Let* $\mathfrak{A}_\mu = \langle A, \sigma, \mu \rangle$ *be a blurry model. Then, if the extended signature* $\sigma_A$ *of this model is at most countable, then the model* $\mathfrak{A}_\mu$ *is a fuzzification of some precedent model.*

Proof. Consider the blurry model $\mathfrak{A}_\mu = \langle A, \sigma, \mu \rangle$ and the sigma-algebra $\Omega(\sigma_A)$. Let us define the mapping $\nu : \Omega(\sigma_A) \to [0; 1]$ as follows:

1. For any sentence $\varphi \in S(\sigma_A)$, let $\nu(K_\varphi) = \mu(\varphi)$.

2. For any class $K \in \Omega(\sigma_A)$ if $\nu(K) = \alpha$, then $\nu(\overline{K}) = 1 - \alpha$.

3. Let $\nu(K_1) = \alpha_1, \nu(K_2) = \alpha_2, ...$ be defined. Then

$$\nu\left(\bigcup_{i \in N} K_i\right) = \lim_{n \to \infty} \nu\left(\bigcup_{i=1}^{n} K_i\right) \text{ and } \nu(\bigcap_{i \in N} K_i) = \lim_{n \to \infty} \nu(\bigcap_{i=1}^{n} K_i).$$

It is not hard to check that the mapping $\nu$ is a probability measure defined on the sigma-algebra $\Omega(\sigma_A)$.

Let $Th(\mathfrak{A}_\mu)$ be the elementary theory of the blurry model $\mathfrak{A}_\mu$. Let us define the class of models

$$E = \bigcap_{\varphi \in Th(\mathfrak{A}_\mu)} K_\varphi.$$

By the condition of the Theorem, the signature $\sigma_A$ is at most countable. Hence, $Th(\mathfrak{A}_\mu)$ is also countable. From this it follows that $E \in \Omega(\sigma_A)$.

Let us show that $\nu(E) = 1$. Let $\varphi_1, \varphi_2 \in Th(\mathfrak{A}_\mu)$. Then, by Proposition 3, we get $\varphi_1 \& \varphi_2 \in Th(\mathfrak{A}_\mu)$. Hence,

$$\nu\left(K_{\varphi_1} \cap K_{\varphi_2}\right) = \nu\left(K_{\varphi_1 \& \varphi_2}\right) = \mu(\varphi_1 \& \varphi_2) = 1.$$

Then, by inductive reasoning, we obtain

$$\nu(E) = \nu\left(\bigcap_{\varphi \in Th(\mathfrak{A}_\mu)} K_\varphi\right) = 1.$$

Let us now show that $Fuz(\mathfrak{A}_E, \nu) \approx \mathfrak{A}_\mu$. In fact, for any sentence $\varphi \in S(\sigma_A)$ we have

$$\frac{\nu(\tau_E(\varphi))}{\nu(E)} = \nu(K_\varphi \cap E) = \nu(K_\varphi) + \nu(E) - \nu(K_\varphi \cup E) = \nu(K_\varphi) = \mu(\varphi).$$

$\square$

## 6 CONCLUSION

Two alternative approaches to formalising object domain knowledge have been considered in this paper. The first approach is based on formalisation on the basis of emperechiic (semantic) knowledge about the object domain. Such formalisation is possible if we know a priori the class of all precedents of the object domain. Then, on the basis of information about the 'significance' of each precedent for a given object domain (collected, for example, by statistical methods), we can construct an evaluation model of the object domain.

The second approach is based on the formalisation of evaluative (subjective) knowledge about the events of the object domain. In this case, formalisation consists in constructing a blurry model of the object domain. The paper shows that if the blurry model is not more than countable, it is possible to reconstruct information about the precedents of the object domain. For the case when the blurry model is more than countable, the question is still open and is the goal of our further research.

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
