# OpenReview forum: "Application of blurry models for semantic modelling of object domains"
_mathai.club/MathAI/2025/Conference — MathAI 2025 Oral_

### Official Review · Reviewer_hwSy · 2025-02-26
**Application of blurry models for semantic modelling of object domains**

**Rating:** 7
**Confidence:** 3

**Review:**

The mathematical definitions and propositions of the article are correct, but there is a lack of an end-to-end example illustrating all the stages of building a formal model of the subject area, including the use of fuzzification and blurring techniques.

There are a small number of typos.

Line 43: knowledge.Ontological

Line 75: denoted

---

### Official Review · Reviewer_cmuH · 2025-02-27
**The article provides the intresting formal description of knowledge representation, accept**

**Rating:** 7
**Confidence:** 3

**Review:**

The article provides the correct mathematical apparatus for reconstructing a set of precedents based on fuzzy models. The main results include a description of the quality of fuzzy models for various object domains. The theoretical basis of the study is a four-level model of knowledge representation, including ontological, theoretical, empirical and statistical levels.

The disadvantages of the article include theoretical abstractness, limited application to non-finite fuzzy models and the lack of empirical data confirming the proposed theoretical conclusions.

---

### Official Review · Reviewer_gwJH · 2025-02-27
**The "Application of blurry models for semantic modelling of object domains" paper can be accepted to the MathAI 2025 conference**

**Rating:** 8
**Confidence:** 3

**Review:**

This paper is devoted to solution of such important task as knowledge representation. Use of blurry models allows authors to solve this task.

This paper has the following disadvantages:
1) Too small review of related works in the "Introduction" section. Thus it is hard to understand relations between considering papers and other works in this research field. For example, authors may consider FASILL programming language (see the "The Fuzzy Logic Programming language FASILL: Design and implementation" paper, DOI: https://doi.org/10.1016/j.ijar.2020.06.002 ) where fuzzy logic has been implemented.
2) Absence of short description of paper structure in the end of "Introduction" section.

---

### Decision · Program_Chairs · 2025-03-08

**Decision:**

Accept (Oral)

**Comment:**

Your article has been accepted and you can give a talk on the article. All articles will be sorted by rating and within the available conference places one author from each article will be invited. If there are not enough places, then you will either have the opportunity to speak remotely or come at your own expense!